# Comprehensive Evaluation of Global Precipitation Measurement Mission (GPM) IMERG Precipitation Products over Mainland China

**Linjiang Nan** [1,2]**, Mingxiang Yang** [1,]*****, Hao Wang** [1]**, Zhenglin Xiang** [3] **and Shaokui Hao** [4]

1   State Key Laboratory of Simulation and Regulation of Water Cycle in River Basin, China, China Institute of Water Resources and Hydropower Research, Beijing 100038, China; nanlinjiang917@163.com (L.N.); wanghao@iwhr.com (H.W.)
2   School of Civil Engineering and Architecture, Guangxi University, Nanning 530004, China
3   China Southern Power Grid, Guangzhou 510630, China; glorioussstar@163.com
4   Tongzhou District People's Government of Beijing Municipality, Beijing 101100, China; zonghejihuake@163.com
*   Correspondence: yangmx@iwhr.com; Tel.: +86-180-4655-5306

**Abstract:** Due to the difficulty involved in obtaining and processing a large amount of data, the spatial distribution of the quality and error structure of satellite precipitation products and the climatic dependence of the error sources have not been studied sufficiently. Eight statistical and detection indicators were used to compare and evaluate the accuracy of the Integrated Multi-Satellite Retrievals for Global Precipitation Measurement Mission (GPM IMERG) precipitation products in China, including IMERG Early, Late, and Final Run. (1) Based on the correlation coefficient between GPM IMERG precipitation products and measured precipitation, the precipitation detection ability is good in eastern China, whereas the root-mean-square error increases from northwest to southeast. (2) Compared with the Early and Late Run, the accuracy of the detection of a light rain of the IMERG Final Run is higher, but the precipitation is overestimated. With the increase in the precipitation intensity, the detection ability weakens, and the precipitation is underestimated. (3) The Final Run has a higher estimation accuracy regarding light rain in western high-altitude areas, whereas the accuracy of the detection of moderate rain, heavy rain, and rainstorms is higher in eastern coastal low-altitude areas. This phenomenon is related to the performance and detection principles of satellites. The altitude and magnitude of the precipitation affect the detection accuracy of the satellite. This study provides guidance for the application of GPM IMERG precipitation products in hydrological research and water resource management in China.

**Keywords:** GPM IMERG; satellite; precipitation; mainland China; accuracy; evaluation

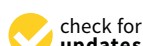

## 1. Introduction

Precipitation connects atmospheric and surface processes and plays a crucial role in the transition cycle of mass and energy. With global climate change, it is important to obtain accurate precipitation data. The accurate measurement of precipitation is important for weather forecasting, disaster monitoring, agricultural production, and scientific research. However, precipitation is one of the most difficult atmospheric variables measured due to its large temporal and spatial variability and abnormal distribution [1,2]. Thus far, precipitation measurement methods have primarily included ground rain gauges, ground-based radar, and satellite remote sensing. Rain gauges often provide relatively accurate precipitation information for a certain area, but the density and spatial patterns of the rain gauge stations diverge significantly across the globe [3,4], either with or without scarce gauge stations over the remote mountainous and oceanic regions [5,6]. Compared with rain gauge data, weather radar has a continuous spatial coverage. Nonetheless, it is quite difficult to establish a worldwide network for weather radar, especially given its

limited utility in snow weather and mountainous terrain [7]. In addition, ground-based radar is vulnerable to electronic signal interferences and is unevenly distributed across land. Therefore, it is difficult to provide large-scale regional and global precipitation data. In contrast, satellite data are characterized by a wide coverage, continuous observation periods, and relatively high spatiotemporal resolutions and have been widely used in hydro-meteorological research in recent years [8]. Nevertheless, due to the indirect retrieval of the precipitation, the satellite precipitation products are inherently subjected to some drawbacks arising from the deficiency of the sensors, retrieval algorithms, and observing frequency [9]. It is therefore necessary and essential to perform an evaluation of the satellite precipitation products before their applications.

Recent research has mainly focused on the comparative evaluation of the Integrated Multi-satellite Retrievals for Global Precipitation Measurement Mission (GPM IMERG) and previously generated Tropical Rainfall Measurement Mission (TRMM 3B42) series products. The effects of the retrieval of TRMM 3B42 and GPM IMERG on extreme precipitation in China from 2000 to 2017 and from 2014 to 2017 were compared and analyzed by Fang et al. [10]. The results showed that TRMM 3B42 products have a limited detection ability regarding extreme rainfall events and that GPM IMERG performs better overall. In addition, the two products performed better in South and East China but worse in high-altitude arid northwestern China. Wang et al. [11] evaluated the performance of TRMM 3B42V7 and GPM IMERG products based on precipitation data obtained at surface observation stations in the northwest. The results showed that IMERG strongly correlates with surface observation precipitation than TRMM 3B42V7 on a multi-temporal scale, but the improvement is not obvious. Yuan et al. [12] compared and analyzed the quality and applicability of the TRMM 3B42V7 and GPM IMERG Final Run at the Yellow River. The results showed that IMERG has a higher accuracy than 3B42V7.

The results of previous studies showed that the GPM IMERG products have been improved to some extent compared with their previous generation. However, the question remains: what is the performance and applicability of GPM IMERG itself and the differences between the Early, Late, and Final Runs? These questions have attracted more attention. Gaona et al. [13] compared the GPM IMERG against gauge-adjusted radar rainfall maps over the land surface of the Netherlands. The results show that the IMERG is a reliable source of precipitation data and can be used as rainfall estimation in a mid-latitude country. Thakur et al. [14] performed an analytical study for the performance of the GPM-IMERG over the Indian and concluded that GPM-IMERG has good quantitative assessment of rain events below 204.5 mm/d, but relatively low assessment for the heavy rainfall events greater than 204.5 mm/d. Anjum et al. [15] evaluated the performance of GPM for northern mountainous region of Pakistan of 2014 to 2016 and reported that the IMERG Final Run performed relatively better. Asong et al. [16] evaluated the performance of IMERG Final Run over different ecological zones of southern Canada, concluding that IMERG overestimated higher monthly precipitation in the pacific maritime ecological zone. Yu et al. [17] studied the performance of GPM IMERG's Early and Late runs by using the correlation coefficient (CC), probability of detection (POD), and other indicators and referring to precipitation data observed at surface rainfall stations in the eastern coastal area of China in 2019. The results show that the accuracy of the detection of heavy rain and rainstorms of the IMERG products is poor. Yang et al. [18] evaluated the inversion accuracy of GPM IMERG for rainstorms and extreme precipitation by using precipitation data recorded at ground stations in the Sichuan Province from 2014 to 2017. The results showed that the precipitation detection ability of IMERG is affected by the rainfall magnitude and terrain. The Final Run overestimates light rain events but has a better general performance. Caracciolo et al. [19] compared ground observation precipitation data of Sardinia and Sicily (Italy) from 2015 to 2016 with IMERG Final Run data to study the reliability of GPM products in complex terrain and geomorphic areas. The results showed that GPM is affected by sea land transition and has a poor accuracy in coastal areas. Su et al. [20] evaluated the applicability of the Early, Late, and Final GPM IMERG

runs in the upper reaches of the Huaihe River from 2014 to 2015 and reported that the three IMERG products overestimate light rain. The Final Run exhibited the best performance. In order to conduct a comprehensive verification of GPM IMERG, it is necessary to further evaluate the performance of different IMERG products over mainland China.

In this study, we comprehensively evaluated the accuracy and applicability of three GPM IMERG products in mainland China using the CC, root-mean-square error (RMSE), relative bias (BIAS), equitable threat score (ETS), POD, false alarm ratio (FAR), critical success index (CSI), and bias (B). The effect of the altitude on the accuracy of the precipitation retrieval of the GPM satellite was analyzed. This study aimed to determine the performance of GPM IMERG products in mainland China, provide references for related research, and lay a foundation for further improvement of the IMERG algorithm.

## 2. Materials and Methods

### 2.1. Study Area

Mainland China is in the eastern part of Asia on the west coast of the Pacific Ocean. The total land area is ~9.6 million $km^2$. China's terrain is high in the west and low in the east and characterized by plains, mountains, hills, basins, plateaus, and other terrain, as well as various climate zones [21]. The complex terrain and landforms lead to diverse climate environments and a variety of temperatures and precipitation types. The spatial distribution of precipitation notably differs in different regions of China; it decreases from the southeastern coast to the northwestern inland. The rainy season in southern China ranges from May to October. It starts early, ends late, and lasts for a long time. However, in northern China, rains are concentrated from July to August and the season is shorter. Precipitation in China mainly occurs in summer and notably differs spatially, with some regions experiencing frequent droughts and others, floods. Therefore, it is important to systematically study the applicability of satellite precipitation data in China.

### 2.2. Data Resources

In this study, two kinds of datasets (rain gauge data and satellite datasets) were used for analyses. The accuracy of the datasets before obtaining have been under quality control. The main work during data processing is to check the "outlier", if the gauges have missing records at certain times, the missing gauges were directly removed to avoid statistical errors.

#### 2.2.1. Rain Gauge Data

In this study, precipitation data observed at ground stations were used, which were provided by the National Meteorological Information Center of the China Meteorological Administration, which have been under rigorous quality control of the source data, and ensured that the actual rate of each element data exceeds 99.9%, and the accuracy of the data is close to 100%. The time resolution of the data from over 800 national meteorological stations was 1 d. Figure 1 shows the topography of China and the distribution of ground observation stations. Ground stations are mainly distributed in low-altitude areas in eastern China. The western region has complex terrain and severe weather. Thus, it is difficult to set up meteorological stations, resulting in a sparse distribution of stations and lack of precipitation data. Overall, the observation stations are unevenly distributed, with larger and smaller numbers in the east and west, respectively. Therefore, it is important to systematically evaluate the accuracy and applicability of satellite precipitation data in China.

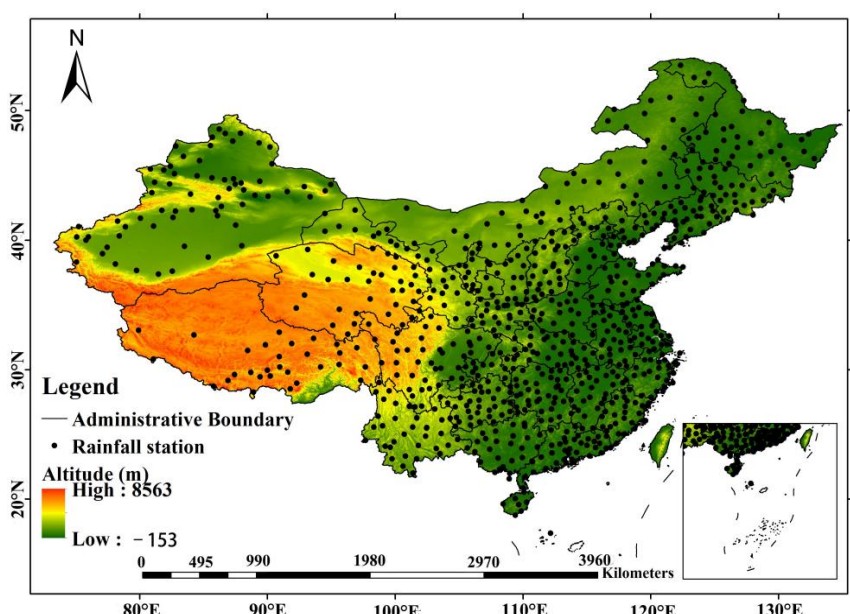

**Figure 1.** Locations of ground observation stations.

2.2.2. Satellite Datasets

The satellite datasets used in this study were obtained from https://disc.gsfc.nasa.gov/ (accessed on 10 January 2021), with a time resolution of 1 day and a spatial resolution of 0.1° × 0.1°, which have been under quality control.

The GPM satellite jointly developed by the National Aeronautics and Space Administration (NASA) and the Japan Aerospace Exploration Agency (JAXA) was successfully launched on 27 February 2014, providing a new generation of precipitation products. The GPM was improved based on TRMM and inherits its mature algorithm and detection technology. The monitoring performance was greatly improved. The spatial and temporal resolutions were improved to enable the obtention of and precipitation data over a larger spatial range [22].

GPM IMERG is widely used product, provides three different run products, Early Run, Late Run, and Final Run. IMERG Early Run and Late Run are quasi-real-time products, with release latency of 4 and 12 h, respectively. IMERG Final Run is a late-stage research product with a delay of 3.5 months. In addition, IMERG Early Run only uses forward propagation, and IMERG Late Run adds backward propagation [23]. The GPM IMERG precipitation data have a high application potential and value in rainfall and disaster prediction. However, there is a lack of research on the accuracy of GPM IMERG satellite data in China. In this study, precipitation data obtained at ground stations were used to compare and evaluate the accuracy of the GPM IMERG Early, Late, and Final Runs from the perspectives of the spatiotemporal distribution, precipitation intensity, and altitude and the temporal and spatial changes of GPM IMERG were analyzed.

*2.3. Study Methods*

Rain gauge data are discontinuous in space, and interpolation processing often brings uncertainty. In order to ensure the accuracy of the rain gauge data, this study compared and evaluated IMERG on the point scale. Satellite data were extracted according to the station coordinates (Equations (1) and (2)) and the accuracy evaluation indexes were used to calculate.

$$\left| Lon_{sat} - Lon_{gro} \right| \leq 0.1 \tag{1}$$

$$\left| Lat_{sat} - Lat_{gro} \right| \leq 0.1 \tag{2}$$

where $Lon_{sat}$ , $Lat_{sat}$ represent the longitude and latitude of the satellite, respectively, and $Lon_{gro}$ , $Lat_{gro}$ represent the longitude and latitude of the ground rain gauge, respectively. 0.1 is the spatial resolution of the satellite.

The accuracy evaluation indexes used in this study can be divided into two categories: statistical and detection indexes. Statistical indicators include the CC, RMSE, and relative bias (BIAS), which were used to evaluate the correlation and deviation between satellite and rain gauge data. Detection indicators include the POD, FAR, CSI, B, and ETS, which were used to evaluate the satellite's ability to capture precipitation events.

### 2.3.1. Statistical Indicators

Based on the data observed at the surface rain gauge, continuous statistical indicators were used to evaluate the GPM IMERG Early Run, Late Run, and Final Run.

1.  Correlation coefficient (CC)

The Pearson CC reflects the strength of the linear relationship between the satellite and rain gauge data. The absolute value ranges from 0 to 1. The closer it is to 1, the more consistent are the satellite and rain gauge datasets, thus the reference value is higher [24].

$$CC = \frac{\sum (X_i - \overline{X})(Y_i - \overline{Y})}{\sqrt{\sum (X_i - \overline{X})^2 (Y_i - \overline{Y})^2}} \tag{3}$$

2.  Root-mean-square error (RMSE)

The RMSE was used to evaluate the deviation between the satellite and rain gauge data. This value is always non-negative. The smaller the value, the smaller is the observation error and vice versa [25].

$$RMSE = \sqrt{\frac{1}{n} \sum_{i=1}^{n} (X_i - Y_i)^2} \tag{4}$$

3.  Relative bias (BIAS)

BIAS refers to the percentage of absolute deviation of the average value, which can measure the deviation between satellite datasets and rain gauge data [26].

$$BIAS = \frac{\sum_{i=1}^{n} (Y_i - X_i)}{\sum_{i=1}^{n} X_i} \times 100 \tag{5}$$

where $n$ is the sample size of the satellite or gauge-based precipitation time series, $X_i$ represents the rain gauge data samples, and $Y_i$ represents satellite precipitation samples.

### 2.3.2. Detection Index

During satellite precipitation inversion, three main types of errors can occur such as missing, false alarm, and hit errors. In this study, the POD, FAR, CSI, B, and ETS were used to evaluate the detection ability of satellite precipitation products.

1.  Probability of detection (POD)

The POD indicates the proportion of correctly detected precipitation events to the total number of events detected by the satellite, which reflects the number of missed precipitation events by the satellite. The POD ranges from 0 to 1. The larger the value, the higher the detection possibility of the precipitation [27].

$$POD = \frac{H}{H + M} \tag{6}$$

2.  False alarm ratio (FAR)

The FAR reflects the proportion of incorrectly detected precipitation events in the total number of events detected by the satellite. This index reflects the degree of false alarms of

the satellite regarding precipitation events, which is also known as the empty alarm rate. The FAR range is [0, 1]. The smaller the value, the lower satellite false alarms [28].

$$\text{FAR} = \frac{F}{H + F} \tag{7}$$

3.     Critical success index (CSI)

The CSI represents the proportion of correctly detected precipitation events to the total number of events recorded by the satellite. It reflects the characteristics of satellite datasets [29].

$$\text{CSI} = \frac{H}{H + M + F} \tag{8}$$

4.     BIAS (B)

The B was used to determine whether the precipitation events were over- or underestimated. The value range is [0, $+\infty$]. A value of B > 1 and B < 1 indicates that the satellite over- and underestimates the precipitation events, respectively [30].

$$B = \frac{H + F}{H + M} \tag{9}$$

5.     Equitable threat score (ETS)

The ETS was used to determine the precipitation detection ability. Its range is [$-1/3$, 1]. The higher the value, the better the detection ability [31].

$$\text{ETS} = \frac{H - H_s}{H + M + F - H_s} \tag{10}$$

$$H_s = \frac{(H + M)(H + F)}{H + M + F + Z} \tag{11}$$

In Equations (4)–(9), H represents the precipitation events successfully captured by ground observation stations and satellites under a specific threshold; M represents the precipitation events captured successfully by ground observation stations but not by satellites; F represents the precipitation events captured by satellites but not by ground observation stations; and Z is the number of events without precipitation based on satellite and ground observation data.

Based on the provisions of the National Meteorological Department on precipitation standards, daily rainfall can be divided into four levels: light rain (<10 mm), moderate rain (10–24.9 mm), heavy rain (25–49.9 mm), and rainstorm ($\geq$50 mm) [32]. To evaluate the ability of the GPM to capture precipitation on a daily scale, four precipitation thresholds of 0.1, 10, 25, and 50 mm/d were selected as the standards, corresponding to "producing precipitation", "light rain", "moderate rain", and "heavy rain", respectively [33].

## 3. Results

### 3.1. Comparative Analysis of Three Types of GPM IMERG Products

In this study, the IMERG Early Run, Late Run, and Final Run were analyzed based on the precipitation data observed at ground stations. The evaluation indexes included the CC, RMSE, B, and ETS.

#### 3.1.1. CC

The spatial distribution of CC from 2014 to 2018 in mainland China is shown in Figure 2. The CC between Early Run and rain gauge data was low in Northwest China, especially in southern Xinjiang and northwestern Qinghai (<0.3). The CC was relatively high in the eastern part of China. The CC in the north of the Hainan Province was larger than 0.6. Figure 2b shows that the improvement of the Late Run was less notable than that

of the Early Run, and the spatial distribution of the CC changed insignificantly. Figure 2c shows the spatial distribution of the CC of the Final Run in mainland China. The overall CC significantly improved and was evenly distributed compared with the Early and Late runs. In addition, the CCs of the southwest and eastern regions were excellent, exceeding 0.5. This phenomenon is related to the density of ground observation stations. The stations in the eastern region of mainland China are relatively dense, while the stations in the western region are sparse, which leads to a high correlation coefficient in the eastern region to a certain extent. In summary, it can be seen that the distribution characteristics of ground observation stations will be affected by topographic conditions, geographical location and other factors, which will further affect the results of CC.

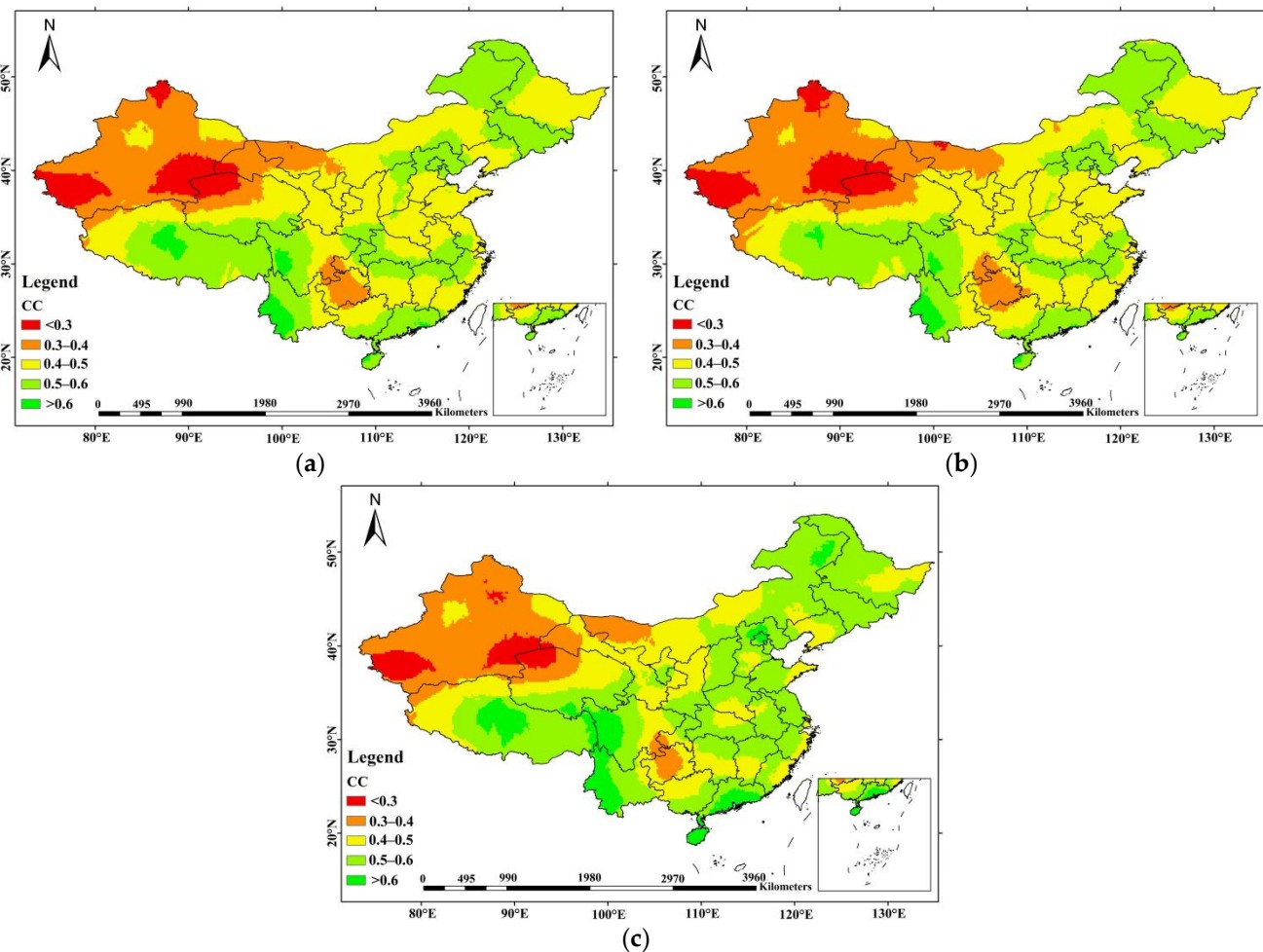

**Figure 2.** Spatial distribution of the CC of the GPM IMERG 2014–2018 data. (**a**) Early, (**b**) Late, and (**c**) Final Run.

In summary, the IMERG Final Run exhibited a higher CC and better performance in mainland China.

### 3.1.2. RMSE

Figure 3 shows the spatial distribution of the RMSE from 2014 to 2018. Overall, the Early, Late, and Final Runs had similar spatial distributions, showing an increasing trend from northwest to southeast and peaks in Guangdong and Guangxi. Compared with the Early Run, the Late Run insignificantly improved, whereas the RMSE of the Final Run was significantly reduced. The maximum value decreased from 11.73 to 9.05, and the low value range significantly expanded in a small area in the southeast. Overall, the RMSE of the southern mountainous area was high, which may be related to the local complex terrain.

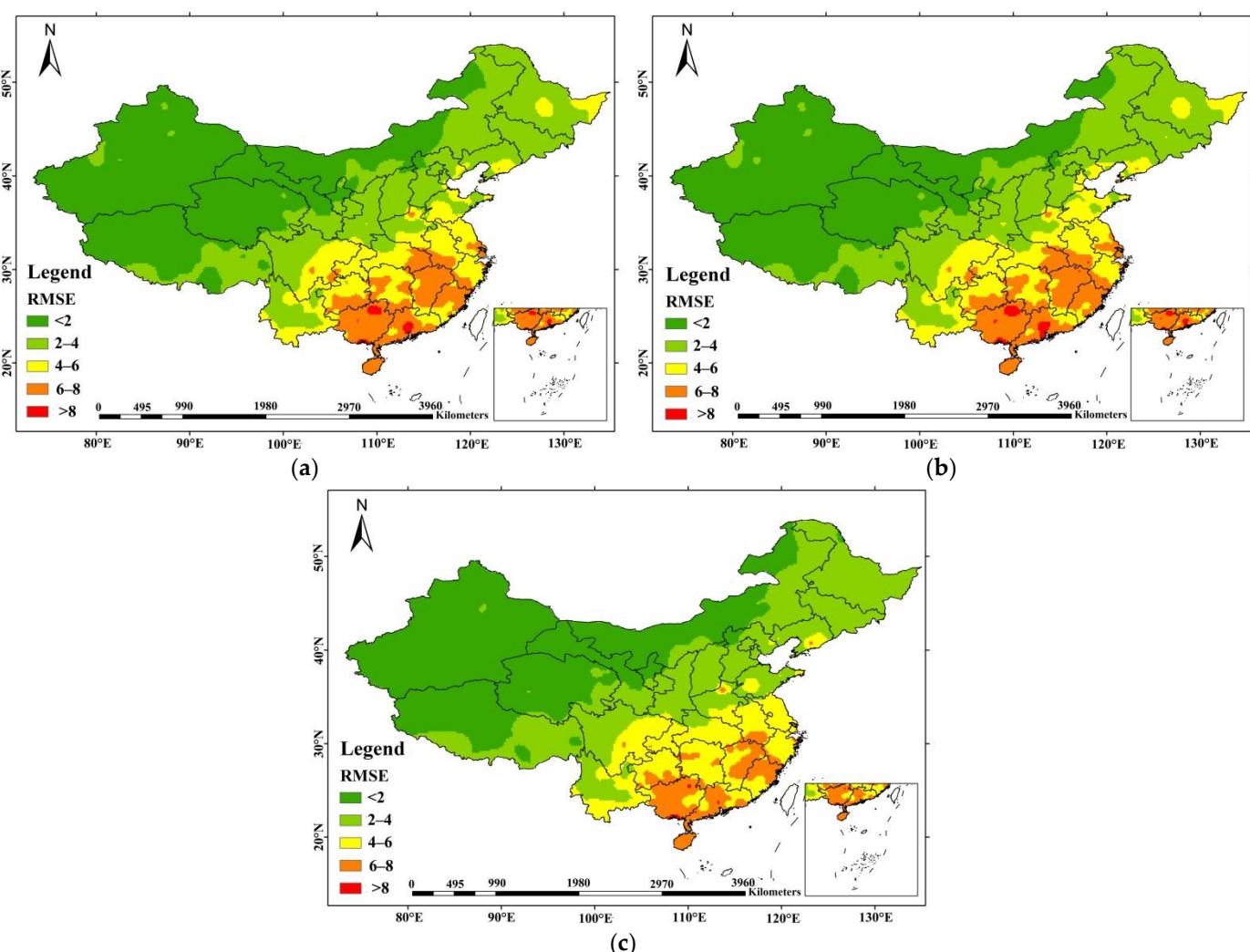

**Figure 3.** Spatial distribution of the RMSE of the GPM IMERG 2014–2018 data. (**a**) Early, (**b**) Late, and (**c**) Final Run.

Satellites generally classify clouds based on the cloud top infrared temperature to distinguish between precipitation and non-precipitation events. Complex terrain conditions in mountainous areas increase the difficulty of satellite detection. Occasionally, the temperature and albedo of rough surfaces in mountainous areas are like the albedo generated by precipitation, which leads to the confusion between rain and non-rain clouds, resulting in the deviation of the precipitation estimation [34].

Thus, in terms of RMSE, the Final Run had the best performance.

### 3.1.3. BIAS

The spatial distribution of B is shown in Figure 4. Based on the comparison of Figure 4a–c, the absolute value of B in most areas was close to zero, indicating an excellent performance. The effect in northwest China was poor, especially in Xinjiang, central Qinghai, and western Inner Mongolia. It is corresponding to the distributions of rain stations over mainland China, dense in the east with better effect and sparse in the west with worse effect. Overall, the Late Run performed insignificantly better than Early Run, whereas the Final Run performed the best. To some extent, it indicated that the Late Run has no significant improvement compared to Early Run.

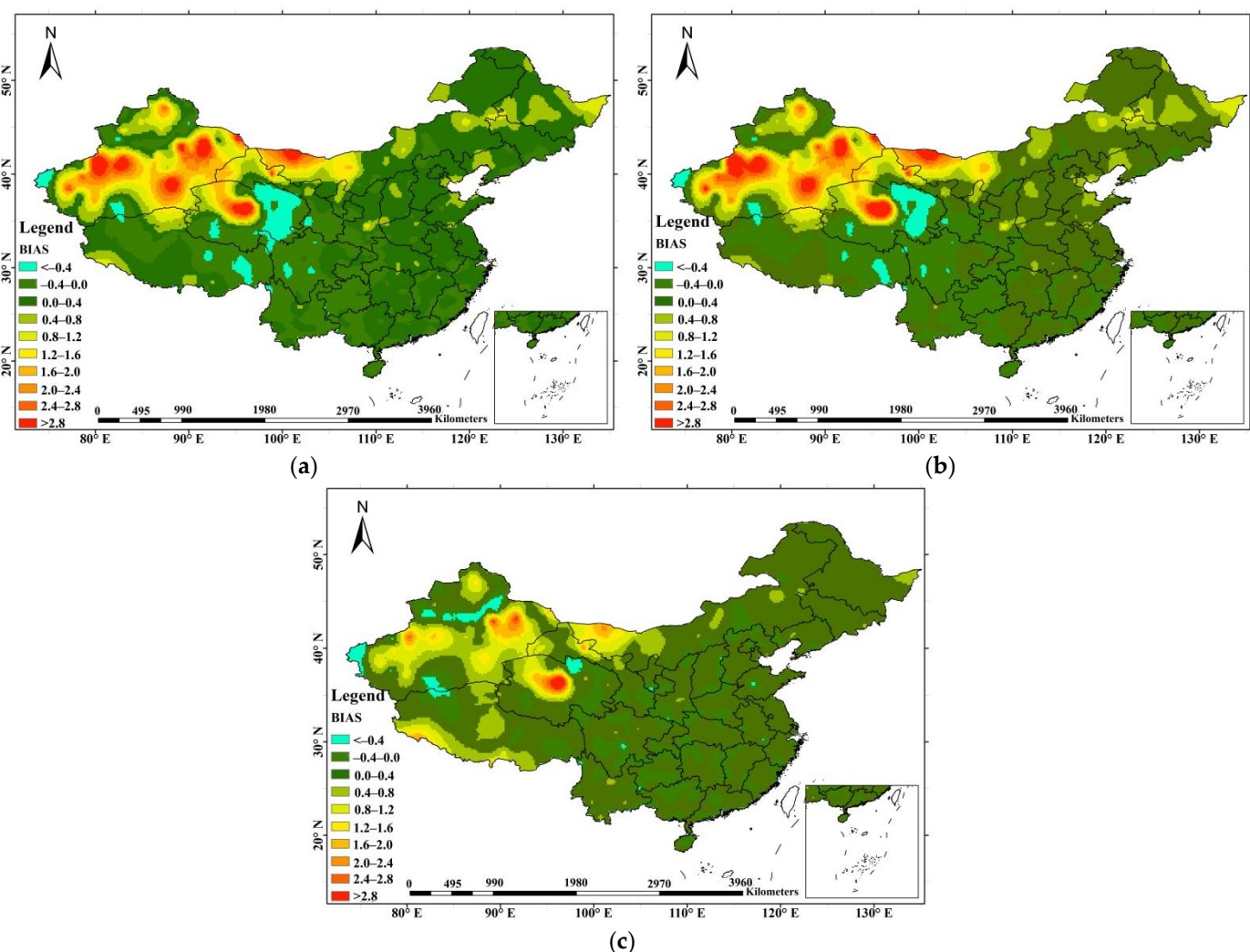

**Figure 4.** Spatial distribution of the BIAS of the GPM IMERG 2014–2018 data. (**a**) Early, (**b**) Late, and (**c**) Final Run.

### 3.1.4. ETS

To further reveal the precipitation detection ability of the GPM, the ETS was used in the study. Figure 5 showed that the detection ability of the Early Run regarding light rain was worse than that based on the threshold of 0.1 mm/d, and the number of areas with an ETS score below 0.1 increased. The detection ability of satellites regarding moderate rain decreased. The performance was better in coastal provinces, and optimal values were obtained in the Guangdong Province. The ETS score of the satellite detection of heavy rain decreased. The highest value was recorded in a small area in eastern Hebei. Regarding rainstorms, the optimal value decreased to 0.1. The optimal value was only obtained in the Hainan Province (the values in other areas were low).

With the increase in the precipitation threshold, the ETS gradually decreased, indicating that the GPM satellite has a better and worse detection ability regarding low-intensity and strong precipitation, respectively. In particular, the GPM better identifies rainfall events with a threshold of 0.1 mm/d. This may be related to the fact that the dual-frequency precipitation radar (DPR) and the GPM microwave imager (GMI) were installed on the GPM satellite to detect weak and solid precipitation.

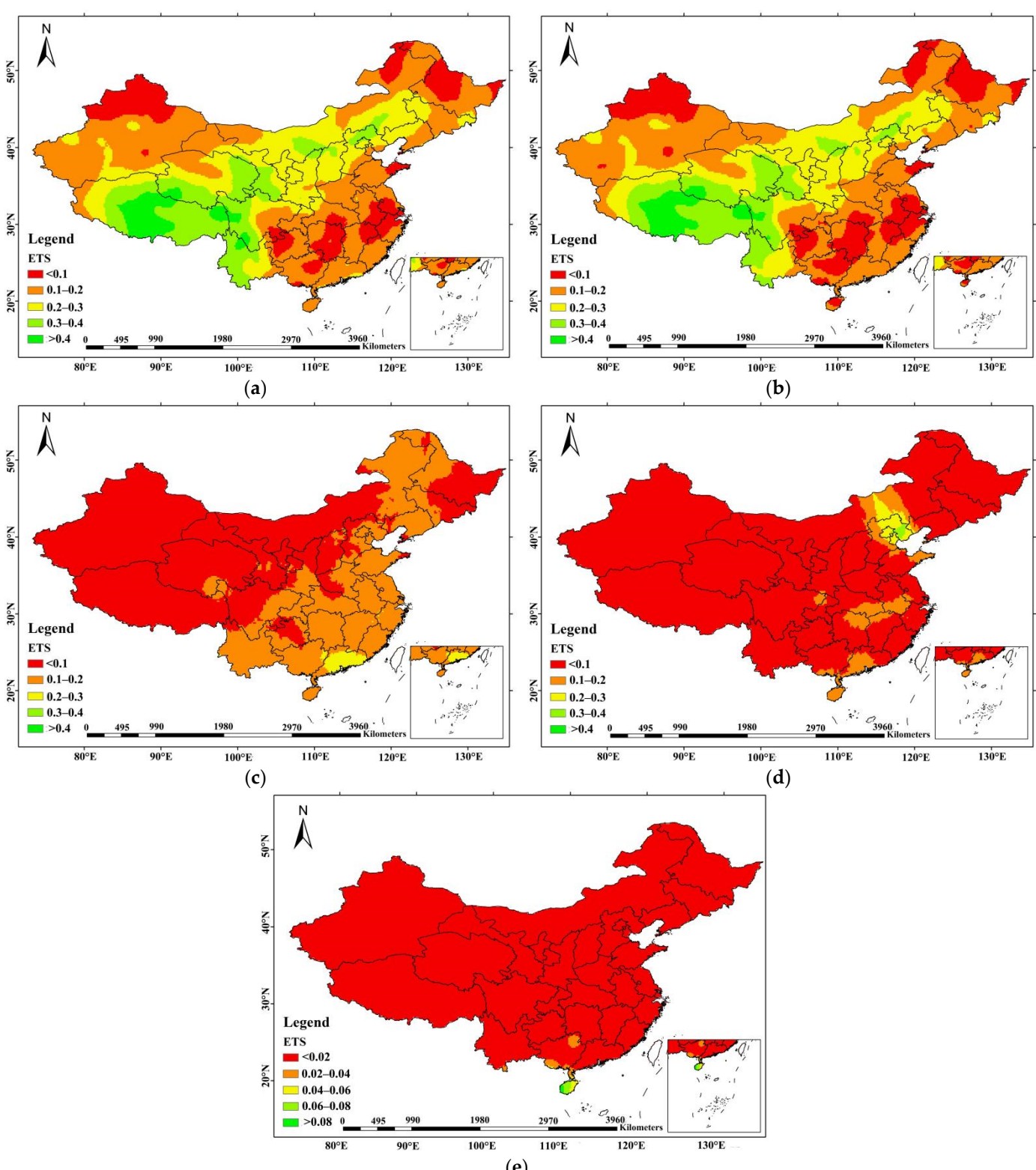

**Figure 5.** Spatial distribution of the ETS of the IMERG Early Run 2014–2018 data. (**a**) Producing rainfall; (**b**) light rain; (**c**) moderate rain; (**d**) heavy rain; (**e**) rainstorm.

Figure 6 describes the spatial distribution of the ETS of the IMERG Late Run. Overall, the detection capability of the Late Run improved compared with that of the Early Run. Based on the threshold of 0.1 mm/d, the IMERG Late and Early runs insignificantly differed. High values were distributed in most areas of Tibet, eastern Yunnan, eastern

Sichuan, Qinghai, and southern Gansu. Regarding light rain, the performances of the Late and Early runs were almost the same. In addition, the detection ability of the Late Run regarding moderate rain improved to a certain extent compared with that of the Early Run. The optimal value was mainly concentrated in Guangdong. It generally performed poorly in the northwest and better in the southeast, showing superior and inferior characteristics in the east and south and west and north, respectively. Furthermore, the distribution of the ETS of the Late Run regarding heavy rain and rainstorms remained unchanged, but the results slightly improved. The performance of the Late Run improved compared to that of the Early Run, but the improvement was relatively small. The score of the ETS indicators was higher for "producing rainfall" and "light rain", whereas it was smaller for moderate rain, heavy rain, and rainstorm, indicating that the GPM satellite's detection of high-intensity precipitation events was inaccurate. It is worth stated that the magnitude of precipitation varies in different time and area, which may influence the strength of the detection capability of the GPM IMERG.

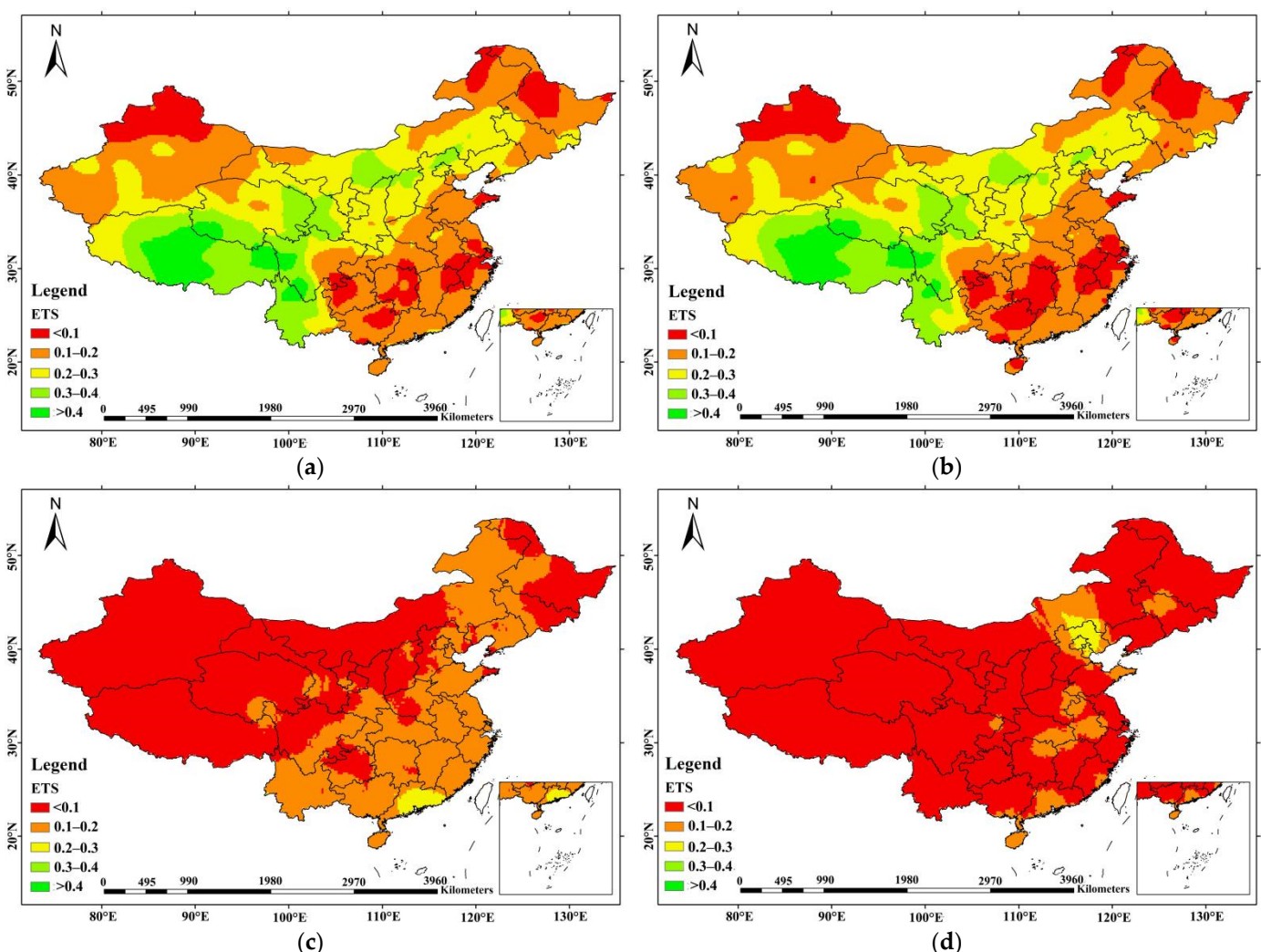

**Figure 6.** *Cont.*

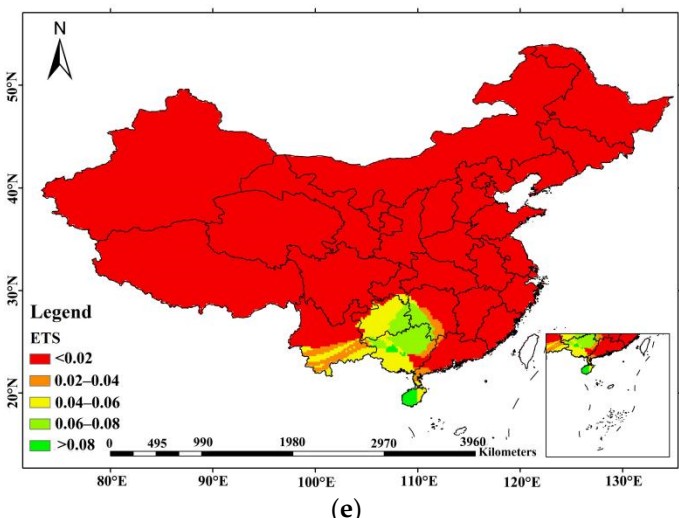

(**e**)

**Figure 6.** Spatial distribution of the ETS of the IMERG Late Run 2014–2018 data. (**a**) Producing rainfall; (**b**) light rain; (**c**) moderate rain; (**d**) heavy rain; and (**e**) rainstorm.

Figure 7 shows the spatial distribution of the ETS of the IMERG Final Run from 2014 to 2018. Overall, the distribution followed that of the Early Run and Late Run. The performance in the eastern region was better than that in the western region under high intensity of rainfall. The ETS value of the Final Run slightly improved compared to the Early and Late runs. It can be seen that there are dependencies on both rainfall intensities and gauge densities and the ETS performs better for light rain. On the one hand, this was related to the installation of the latest dual-frequency radar system on the GPM satellite and improvement of the performance of the microwave radiometer, both of which enhance the detection of weak and solid precipitation. In particular, the precipitation sensitivity of spaceborne radar increased from 0.5 mm/h for PR to 0.2 mm/h for DPR. The performance of SPPs in light rainfall events in GPM era has attracted wide attention of algorithm developers and data users. Although the rainfall radar DPR loaded on the GPM core observation platform enhances the detection of solid and micro precipitation, the detection accuracy of large and medium-sized precipitation has not been further improved. On the other hand, with the increased gauge densities, the performance of the IMERG rainfall product in estimating values of light rainfall has been significantly improved.

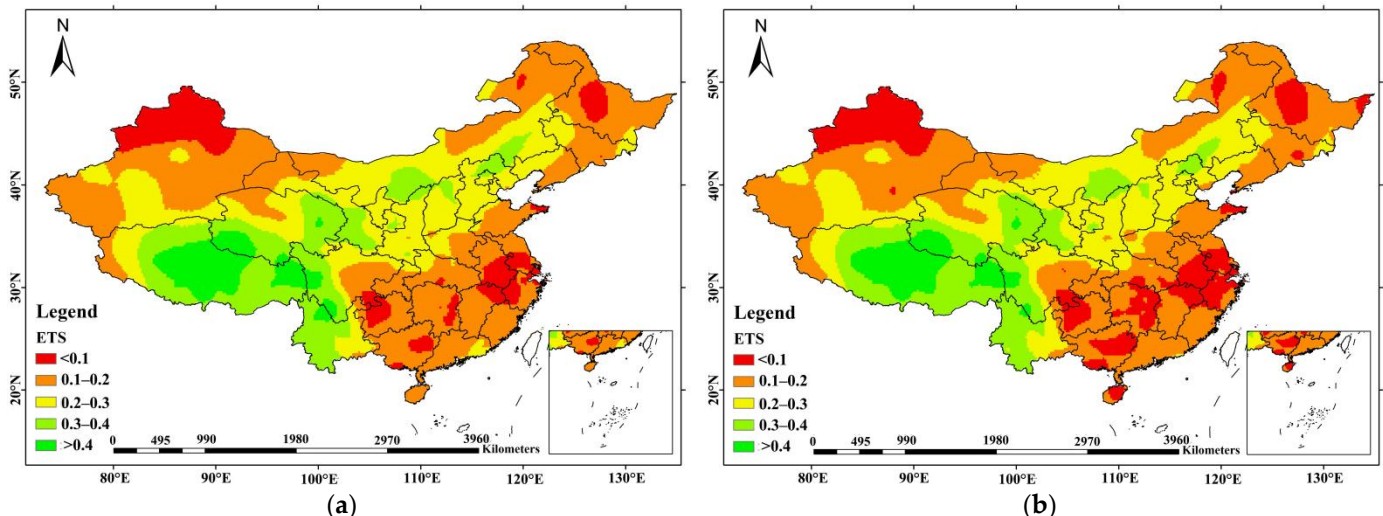

(**a**)                                                  (**b**)

**Figure 7.** *Cont.*

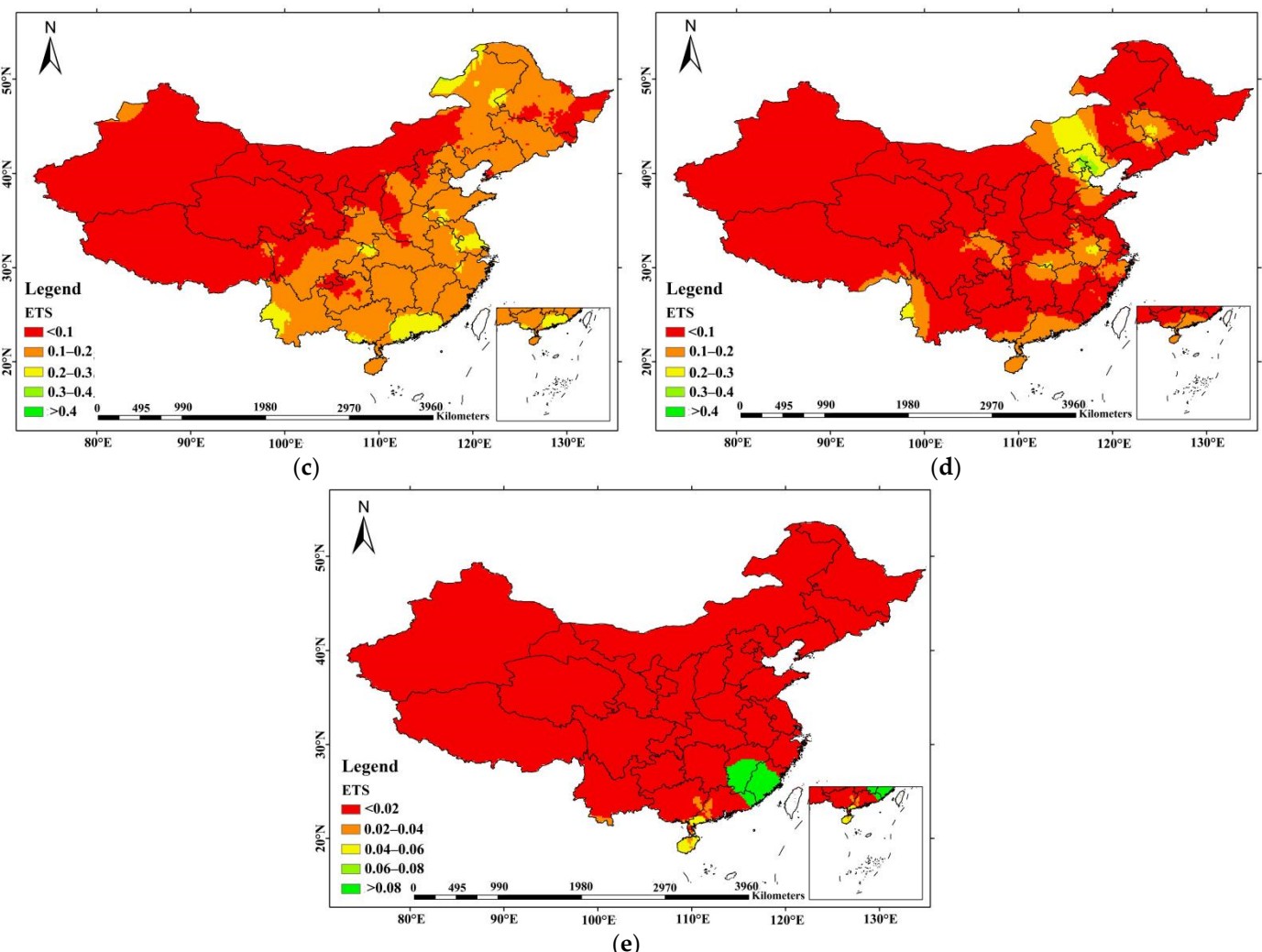

**Figure 7.** Spatial distribution of the ETS of the IMERG Final Run 2014–2018 data. (**a**) Producing rainfall; (**b**) light rain; (**c**) moderate rain; (**d**) heavy rain; and (**e**) rainstorm.

### 3.2. Deeper Evaluation of the Final Run

Based on the comprehensive evaluation of the GPM IMERG Early, Late, and Final Runs and all indicators, the Final Run performed better than the other runs. To further evaluate the performance of the Final Run, the POD, FAR, CSI, and B were used.

#### 3.2.1. POD

Based on the rain gauge data from 2014 to 2018, the spatial distribution of the POD was analyzed in this study for different precipitation magnitudes. Figure 8a shows that the POD of the Final Run in northwest China was lower, especially in Xinjiang, whereas it was higher in the central and eastern regions, especially in the southeastern coastal district for which a wide range of high values (>0.9) was obtained. The POD shown in Figure 8b was smaller than that in Figure 8a. The POD of light rain by the satellite was lower than that under the threshold of 0.1 mm/d. Figure 8c shows the distribution of the Final Run's POD of moderate rain. A high value (indicating an excellent performance) was observed only in the southeastern region, whereas the POD was low in other areas, especially in Xinjiang, Tibet, Qinghai, Gansu, and Ningxia, with values below 0.2. In terms of heavy rain, the overall performance of the Final Run in the country was poor. The number of high-value areas, such as eastern Qinghai, Gansu, eastern Hebei, Beijing, Tianjin, Guangdong, Hainan,

and other regions, was small. This result is similar to the study of Li et al. [35], but the results have been extended by evaluating various precipitation magnitudes.

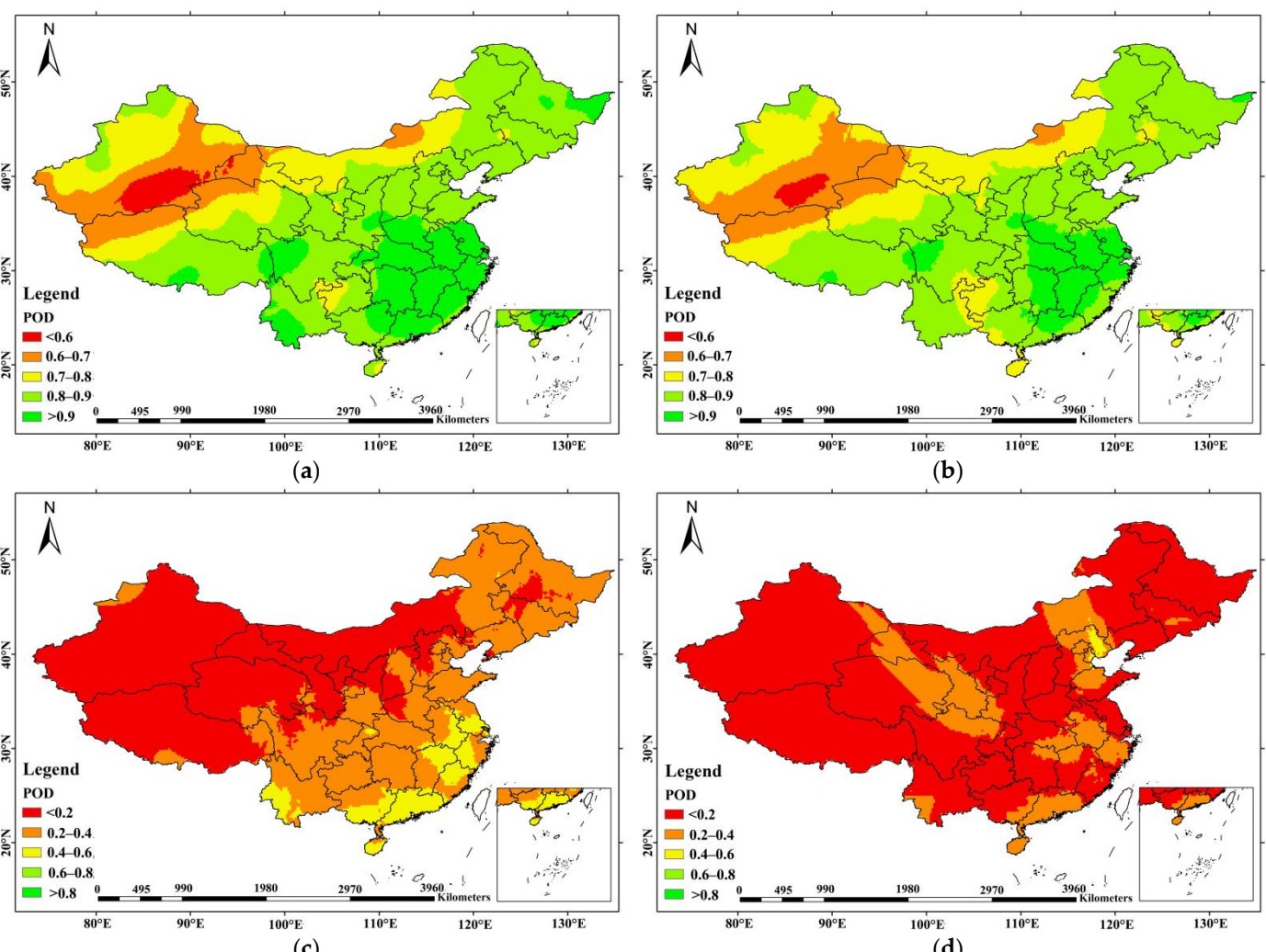

**Figure 8.** Spatial distribution of the POD of the IMERG Final Run 2014–2018 data. (**a**) Producing rainfall; (**b**) light rain; (**c**) moderate rain; (**d**) heavy rain.

### 3.2.2. FAR

The FAR of the Final Run was good in the east but bad in the west. With "producing rainfall" and "light rain", the FAR was better in most eastern regions, with values below 0.4. The POD in the western region was lower than that obtained in the southeast of Xinjiang, whereas the FAR in the northwest of Xinjiang was low and the POD was high, which may be related to the Yili River Valley. Yili Valley is surrounded by mountains on three sides and forms an angular valley to the west. Under the influence of water vapor from the Atlantic Ocean, warm and humid air flows upward along the windward slope, helping to generate precipitation. Most of the precipitation accumulates in Yili Valley, which to a certain extent reduces the effect of drought on the detection ability in this region [36]. Figure 9c,d show that the detection ability of high-intensity precipitation of the Final Run was poor and there is room for improvement. With moderate rain, low FAR values were mainly distributed in the southeast, whereas values above 0.7 (with large errors) were observed in other areas. With heavy rain, the FAR was below 0.6 in Eastern Hebei, Beijing, Tianjin, and Hainan, but high in most of the other regions.

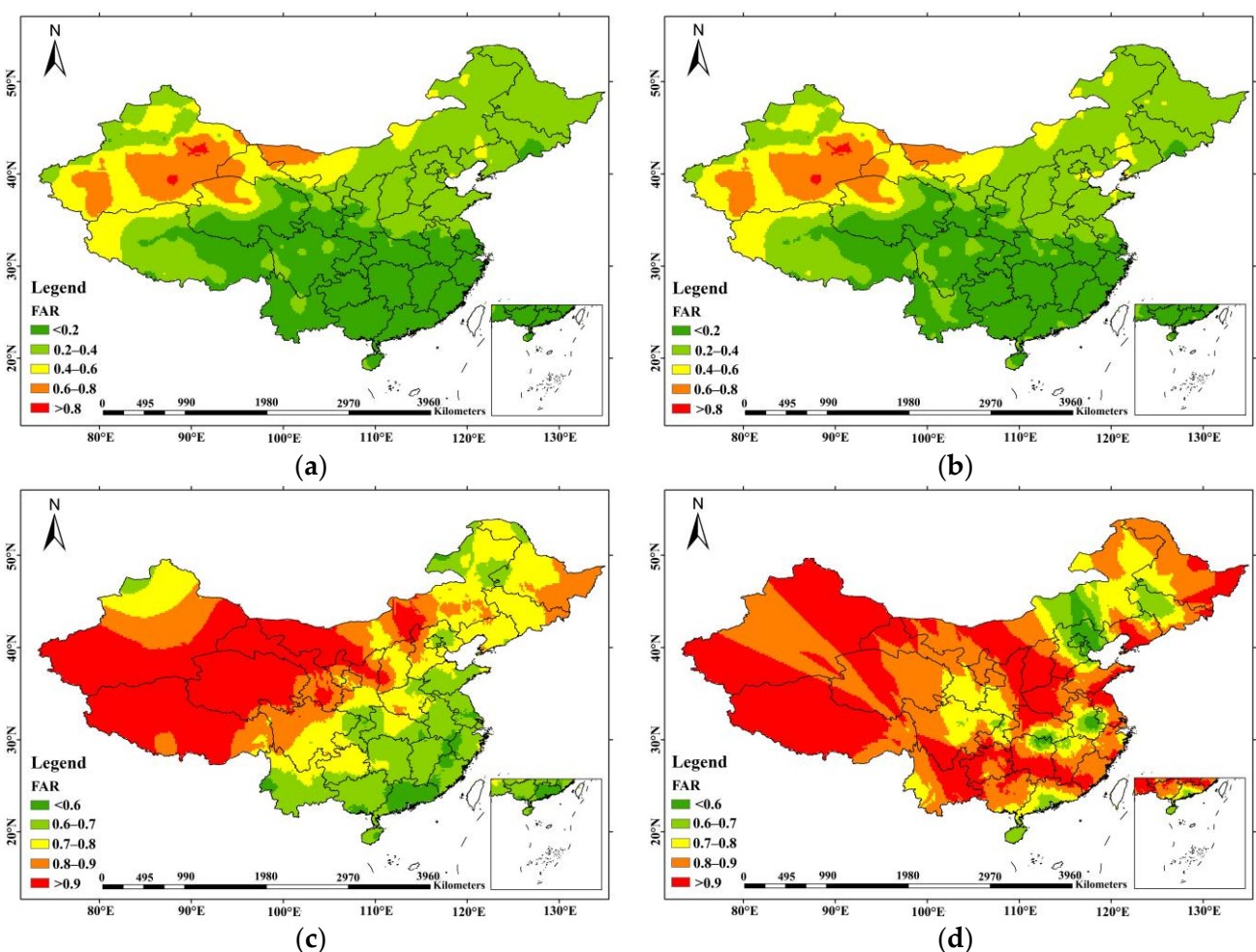

**Figure 9.** Spatial distribution of the FAR of the IMERG Final Run 2014–2018 data. (**a**) Producing rainfall; (**b**) light rain; (**c**) moderate rain; (**d**) heavy rain.

### 3.2.3. CSI

The CSI reflects the POD and FAR performances. It can further reflect the detection capability of the GPM. The CSI value of the Final Run in Figure 10 shows that the performance in the east was better than in the west and the worst in the northwestern region, indicating that it is difficult for satellites to accurately detect precipitation in arid areas, partly since raindrops evaporate before reaching the ground. The performance in the southern region was superior to that in the northern region, which shows that the precipitation products of satellites perform better at low latitudes than at high latitudes. This is reasonable due to the fact that low-latitude regions generally receive more rainfall than the other regions and the topography is relatively flat, which is favorable for precipitation estimations. However, compared with other areas in the south, the satellite detection ability near the Sichuan Basin was poor, which may be due to its location in the east of the Qinghai-Tibet Plateau and the influence of the Indian and East Asia monsoons and atmospheric circulation system. The terrain and climate types were complex, especially in several areas of the western Sichuan Province in which rain and extreme precipitation are easily generated, which affects the detection capability of the satellite to a certain extent. The accuracy difference of satellite precipitation products in different regions of Sichuan is obviously related to its special geographical location and the limitations of satellite sensors. ① First of all, Sichuan is located in the east of the Qinghai Tibet Plateau. At the same time, affected by the Indian and East Asian monsoon and the Plateau Atmospheric circulation system, the terrain and climate types are extremely complex. In particular, topographic and convective rain are

easy to form in some parts of the west, which interferes with the detection of satellite sensors to a certain extent. However, the terrain of Sichuan Basin is relatively flat with few factors affecting the detection accuracy of satellite sensors. ② Sichuan is an area of frequent extreme precipitation in summer. The detection ability of microwave and infrared sensors for trace and heavy precipitation is poor, which makes the detection ability of satellite precipitation products for different magnitudes of precipitation various [37].

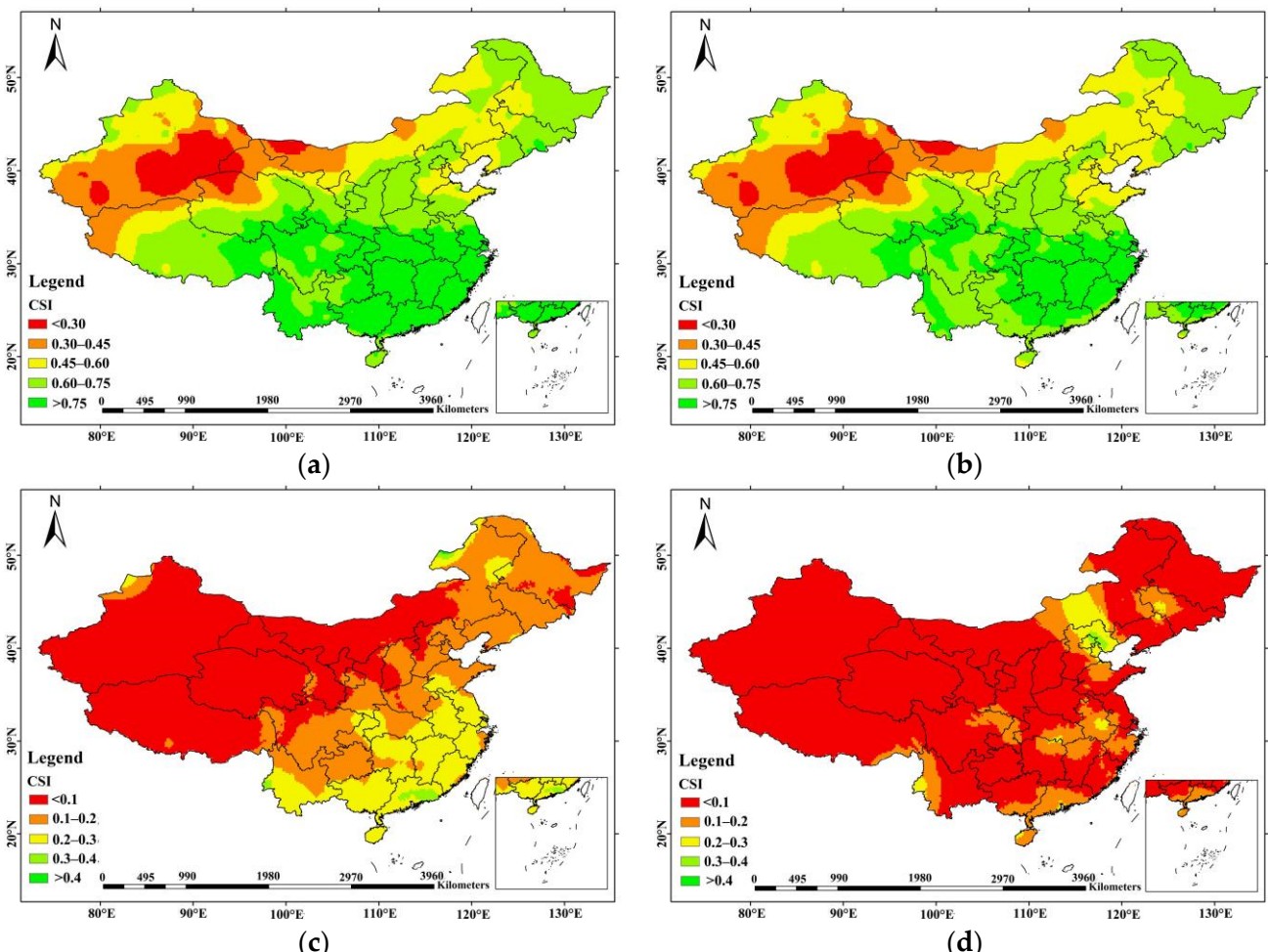

**Figure 10.** Spatial distribution of the CSI of the IMERG Final Run 2014–2018 data. (**a**) Producing rainfall; (**b**) light rain; (**c**) moderate rain; (**d**) heavy rain.

In addition, with the increase in the magnitude of precipitation, the CSI worsened.

### 3.2.4. BIAS

Figure 11 shows that the Final Run overestimated light rain in most areas but underestimated moderate and heavy rain. This result is consistent with others' conclusions [38,39].

Thus, the B of the Final Run was affected by the precipitation intensity. It generally overestimated low-intensity precipitation and underestimated the high-intensity precipitation.

### 3.3. *Effect of the Altitude on the Precipitation Inversion Accuracy*

Topography is not only an important factor influencing the climate system, but also the dominant factor affecting the local microclimate (especially precipitation). The complex topography of mainland China may cause errors in the precipitation retrieval due to changes in the altitude. Therefore, the effects of orography and diverse topography cannot be ignored, and need further investigation.

Based on rain gauge data, the accuracy and applicability of the IMERG Early, Late, and Final Runs in China were evaluated in this study. The results showed that the Final Run performed the best. Based on in-depth analysis of the Final Run, the detection capability correlated with the altitude. The results showed that the spatial distribution of the ETS of various magnitudes at different elevations revealed the effect of the terrain on the satellite detection ability.

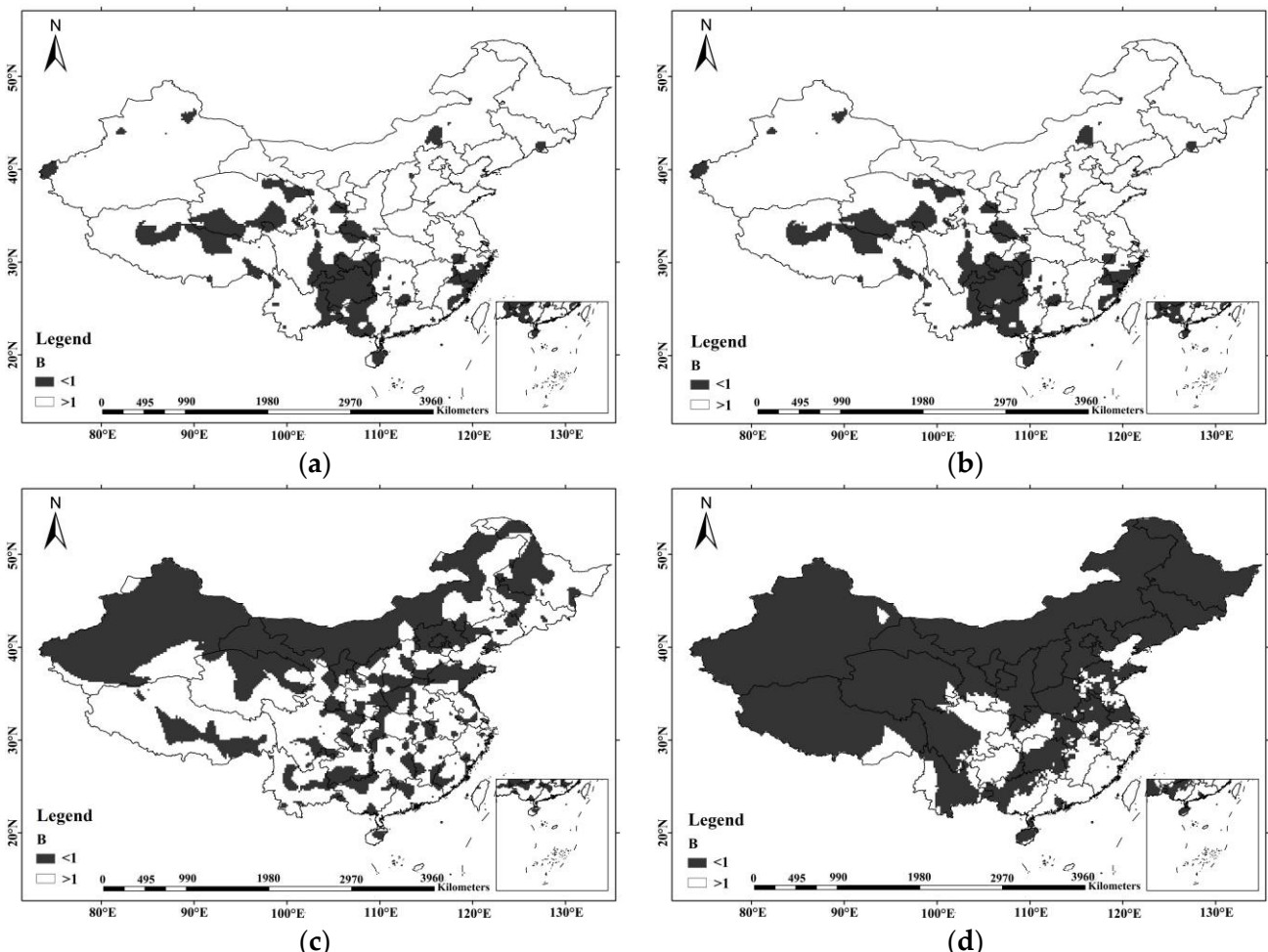

**Figure 11.** Spatial distribution of the BIAS of the IMERG Final Run 2014–2018 data. (**a**) Producing rainfall; (**b**) light rain; (**c**) moderate rain; (**d**) heavy rain.

Figure 12 shows that the ETS score of high-altitude areas in western China was generally high and the values for the "producing rainfall" and "light rain" were optimal. With the decrease in the altitude, the accuracy of the Final Run decreased. The estimation ability of the Final Run was excellent in high-altitude areas and poor in low-altitude areas. Regarding moderate and heavy rain, the Final Run performed better in the east and worse in the west. Optimal values were obtained in the eastern coastal plain areas. The results further shows that the different land cover, geographical position of rain gauge, elevation, wand winds speed and direction could influence the amount of precipitation and hence result in variation of the ETS from north to south [40].

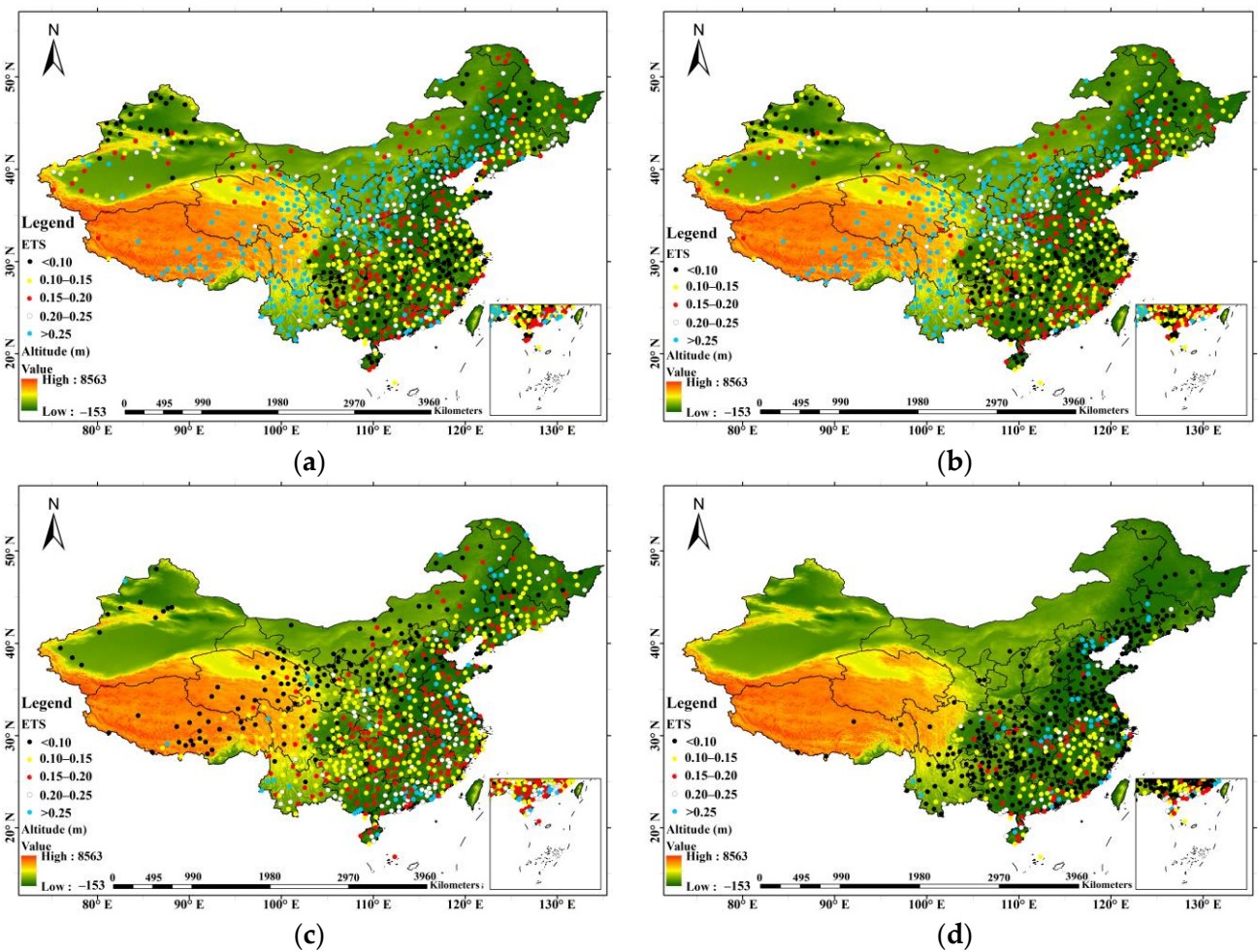

**Figure 12.** Spatial distribution of the ETS of the IMERG Final Run 2014–2018 data. (**a**) Producing rainfall; (**b**) light rain; (**c**) moderate rain; (**d**) heavy rain.

## 4. Discussion

It can be seen that IMERG products tend to underestimate precipitation and fail to perform well for high-intensity precipitation according to the evaluation indices. In addition, the overall performance of IMERG Late Run is close to that of IMERG Early Run, although IMERG Late Run has higher latency and more processing. On the whole, the near-real-time IMERG Early Run products show comparable hydrology performance as the IMERG Late Run product, presenting the great potential for the realtime application, however the IMERG Final Run performs the best.

To further study the post-real-time IMERG Final Run, more indices were introduced. Based on the comparison of Figure 11a–d, precipitation in southwestern mountainous areas was underestimated, which indicates that IMERG may not be effective in estimating topographic precipitation. The results showed that using satellites with infrared (IR) and passive microwave (PMW) sensors for the estimation of precipitation was ineffective in detecting precipitation in mountainous areas. The reasons IMERG underestimated precipitation in mountainous areas may include these aspects: (1) satellites generally assume that heavy precipitation is caused by deep clouds, but the topographic precipitation of shallow clouds is ignored; (2) the "brightness temperature" of warm terrain clouds generally exceeds the IR threshold and is difficult to capture by satellites; (3) the content of ice crystals in warm terrain clouds is generally too low to be detected by PMW sensors; (4) mountainous areas are vulnerable to monsoon, which increases the complexity of precipitation; (5) there is topographic rain in mountainous areas, which is mainly manifested in more

rain on windward slope and less rain on leeward slope; and (6) the complex terrain and few observation stations in high-altitude mountainous areas lead to the underestimation of precipitation by GPM IMERG products.

Mainland China contains various climates and elevation bands, and the IMERG products might perform quite differently. In this case, a primary exploration focusing on the altitude factor is desperately needed. The analysis of the spatial distribution of the ETS of the IMERG Final Run can be performed from based on its relation to the performance of the satellite. The GPM satellite has the latest DPR and Ka-band radar (kaPR). Research showed that the operating frequency of the kaPR is 35.5 GHz, the sensitivity is higher, and ice particles can be observed, which enhances the detection ability of solid precipitation of the GPM. The climate in high-altitude areas of western China is cold and the perennial precipitation is mainly snow and light rain, which can be easily detected by the GPM satellite [41]. The ETS reflects the detection capability of the satellite. Therefore, the ETS of light rain in the western high-altitude area was excellent. Compared with the western high-altitude areas, the eastern coastal districts are characterized by lower altitudes and the precipitation is mainly moderate and accompanied by heavy rain. This is conducive to the accurate detection of local moderate and heavy rain by satellites. However, it is also related to the evaporative law in rain spell. The satellite detects precipitation information on the top cloud by using microwave and infrared sensors. The estimated precipitation experiences evaporation and air resistance dissipation before reaching the ground [42]. Due to the small magnitude of light rain, evaporation occurs before the rain reaches the ground from the top of the atmosphere. Evaporation is positively correlated with the distance between the atmosphere and ground. The distance between the top layer of the atmosphere and ground in the western high-altitude area is small and the influence of evaporation is weak. Therefore, the detection ability of the satellite is high. Owing to the large distance between the top layer of the atmosphere and the ground in the eastern low-altitude area, light rain easily evaporates. Therefore, the ETS score was low. When the precipitation magnitude increases, the effect of evaporation is relatively weakened. Here, the eastern coastal low-altitude areas with medium and high-intensity rainfall are less affected by evaporation and the ETS score was relatively high.

## 5. Conclusions

In-depth accuracy evaluation and applicability analysis of satellite precipitation products is not only an important way to reveal errors in satellite datasets, but also necessary for precipitation data. Based on the daily rain gauge data in China from 2014 to 2018, the GPM IMERG Early, Late, and Final Runs were evaluated in this study. The main conclusions are:

1.  Based on the evaluation of the IMERG Early, Late, and Final Runs using the CC, RMSE, BIAS, and ETS, the CC, and BIAS values of the products in eastern China are better than those in western China, whereas the RMSE increases from northwest to southeast, with a peak in Guangdong, Guangxi, and Hainan. Regarding the high latitudes in Northeast China, the IMERG products show a high correlation and low deviation in several areas. Therefore, IMERG products can be used in China's high-latitude areas, but they are inferior to non-real-time applications (such as water resources management and hydrology modeling) at low latitudes. The IMERG Early and Late Run (near real-time products) can be used for real-time applications such as flood forecasting in low-latitude subtropical humid areas in southeast China. In high-latitude areas, the applicability of near real-time Early and Late run products is low, but they have a real-time application potential and must be further explored. The reasons for the poor performance in northwest China are the complex terrain, arid climate conditions, and significant interference of wind and snow regarding satellite precipitation inversion, resulting in uncertainty in the evaluation results. In addition, although the IMERG Late Run contains more remote sensing information and has a longer delay, its performance is only slightly better than the IMERG Early Run.

Therefore, it is recommended to first consider the IMERG Early Run product with a shorter delay time (~4 h).

2.  When using the ETS index to evaluate the accuracy of satellite products under various levels of precipitation, better results are obtained for "producing rainfall" and "light rain" in most areas of Tibet, Eastern Yunnan, Eastern Sichuan, Qinghai, and southern Gansu. With the increase in the precipitation magnitude, the evaluation results gradually worsen. Regarding the ETS scores of moderate rain, heavy rain, and rainstorm, high values can be observed in the eastern and southern coastal areas. Overall, except for individual regions, the performance of the Late Run is slightly better than the Early Run. The improvement of the Final Run is the most notable.

3.  POD, FAR, CSI, and B were used to analyze the estimation performance of the Final Run. The results show that the performance in South and East China with a humid monsoon climate is better, whereas it is worse in the western drought region with a high altitude. On the one hand, topographical and climatic conditions significantly affect the satellite estimation results. However, the performance is related to the algorithm of the Final Run, which aims to correct the deviation between the GPM satellite detection results and the GPCC precipitation datasets. Therefore, when the Final Run product is used for the evaluation, the results are also affected by the number of ground reference stations. In addition, the Final Run has a strong ability to detect light rain, but it will overestimate it. This may be summarized as the following reasons: (1) the over-correction of the satellite inversion algorithm, which misinterprets several non-rainfall events as light rain events; and (2) light rain is easy to be missed at ground observation stations, which will affect the evaluation results. With the increase in precipitation, the detection ability decreases and underestimation occurs. Generally, it is necessary to improve the detection ability of the Final Run for heavy rain and rainstorms and strengthen the detection of precipitation in the western region.

4.  Based on the overall performance of the Final Run, the evaluation of the eastern region was better than those of the western region. To reveal the reason for this difference, the effect of the altitude was studied. Based on the results, the performance of the Final Run is excellent regarding the detection of light rain in high-altitude areas, but it is poor in low-altitude regions. Regarding moderate and heavy rain, the performance of the Final Run is better in low-altitude areas in the east and poor in the west. Optimal values were mainly obtained in the eastern coastal areas. The results are related to the principle of precipitation detection by the satellite. The satellite detects precipitation information on the cloud top by using microwave and infrared sensors. The estimated precipitation experiences evaporation and air resistance dissipation before reaching the ground. As the western high-altitude area is close to the cloud top, the loss is smaller when light rain is detected, and the detection result is relatively accurate. Evaporation insignificantly affects moderate and heavy rain; a large amount of precipitation occurs in the eastern region. Based on various factors, the satellite has a better detection ability in the eastern and southern coastal plain areas.

5.  The comparison of various indicators shows that IMERG products perform poorly in northwest China, which may be related to local climate conditions. The northwestern region is relatively arid, and evaporation is strong. Not all liquid water observed in the atmospheric profile represents precipitation. The precipitation retrieved by the satellite is based on the cloud structure and its algorithm and model may not fully consider the evaporation level in these areas. In addition, due to the strong evaporation in arid areas, parts of the liquid water evaporate during landing, leading to errors in the estimation of precipitation in arid areas.

The factors affecting the performance of GPM IMERG products mainly include the geographical location, topographic conditions, precipitation intensity, and regional meteorological station density. The results show that the performance of IMERG is relatively poor

in arid areas with complex terrain and high altitude. Therefore, the GPM algorithm must be improved for the application in areas with complex terrain and sparse observation data.

Considering the present study and the previous research mentioned, the following prospective goals should be considered.

1.  Based on this research, the detection ability of light rain of GPM IMERG precipitation products in western high-altitude areas is better than that in most eastern areas, but contrasts that of moderate and heavy rain. However, the detection ability has not been evaluated on a seasonal scale. Such an evaluation should be conducted in future research to better understand the characteristics of GPM products towards various seasons.
2.  In this study, daily precipitation data were analyzed. The GPM IMERG 30 min precipitation data should be evaluated in future research with more indices, in order to get more comprehensive results.
3.  The evaluation results at the national scale may not apply to the watershed scale, such as multi-terrain rainfall and extreme precipitation in the Sichuan Basin. Therefore, it is necessary to conduct more in-depth research on watershed division and altitude factors, especially the districts with unique topographic conditions.

In future research, precipitation products should be expanded, and multi-scale comparative evaluation should be conducted [43]. In addition, data fusion should be considered to obtain a set of precipitation data with stronger applicability and higher accuracy.

**Author Contributions:** Conceptualization, L.N. and M.Y.; methodology, L.N.; software, L.N.; validation, L.N., M.Y., H.W., Z.X. and S.H.; formal analysis, L.N.; investigation, L.N.; resources, L.N.; data curation, L.N.; writing—original draft preparation, L.N. and M.Y.; writing—review and editing, L.N.; visualization, L.N.; supervision, M.Y.; project administration, M.Y., H.W., Z.X. and S.H.; funding acquisition, M.Y., H.W., Z.X. and S.H. All authors have read and agreed to the published version of the manuscript.

**Funding:** This research was supported by the project of State Key Laboratory of Simulation and Regulation of Water Cycle in River Basin (SKL2020TS01), National Natural Science Foundation of China (U1865102) and the project of China Southern Power Grid (0200002019030304SG00003).

**Acknowledgments:** This work was supported by the project of State Key Laboratory of Simulation and Regulation of Water Cycle in River Basin (SKL2020TS01), National Natural Science Foundation of China (U1865102) and the project of China Southern Power Grid (0200002019030304SG00003). We are grateful to the scientists on the NASA science team for providing satellite precipitation and DEM data. We thank the China Meteorological Data Service Center for providing gauge-observed precipitation data.

**Conflicts of Interest:** The authors declare no conflict of interest.

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
