# Peer review of "Comprehensive Evaluation of Global Precipitation Measurement Mission (GPM) IMERG Precipitation Products over Mainland China"

_water, doi:10.3390/w13233381_

Round 1

Reviewer 1 Report

This paper evaluated the accuracy of GPM satellite product in China. The topic is interesting and important for the hydrology science. I have some concerns for this manuscript which are shown below:

  1. For section 2.3. Why these indicators are selected?
  2. Section 3, I suggest to separate the results and discussion part. In the discussion part, the results of the different index need to be compared and more guidance for the hydrological applications for the Early and late Run of GPM data need to be analyzed.
  3. The occurrence for heavy rain and rainstorm are quite different in west and east China. Do you consider the effect of this difference on your results?
  4. You need to compare your results to previous studies to show the new findings of your study. For example:Zhang, L. , Li, X. , Cao, Y. , Nan, Z. , & Yu, W. . (2019). Evaluation and integration of the top-down and bottom-up satellite precipitation products over mainland china. Journal of Hydrology.

Author Response

Thank you for your comments concerning our manuscript “Comprehensive evaluation of Global Precipitation Measurement Mission (GPM) IMERG precipitation products over mainland China”, that we submitted to water (water-1447762).

We found the reviewers’ comments to be very valuable and helpful in improving our presentation, as well as important for guiding significantly to our research. We have read the comments carefully and the manuscript have been thoroughly rechecked and provided in the revised manuscript (red highlighted) according to these comments and suggestion. Attached please find our revised manuscript, and listed below are our point-by-point responses to the reviewer suggestions, which are highlighted in red. Thank you again for handling this manuscript.

With all best wishes, 

                         Yours sincerely

Linjiang Nan

Reviewer 2 Report

With a series of scores and 4-year data set, this study examines the performance of GPM IMERG precipitation products over mainland China. In addition, the products of early, late and final run are further examined. This study is important for both research and operation unit, and it deserve to be published in the journal. However, there are some questions and issues need to be addressed. Here are my comments and suggestions:

Major: 

  1. The introduction: authors over emphasized the power of satellite data and only state the advantage of satellite data, and underlined the disadvantage of ground-based data. If this was true, why it is necessary to validate the GPM IMERG data with ground-based observations? Furthermore, the current literature reviews are highly focused on previous study over mainland China, readers would like to know the progress around the world since GPM covers a very wide area and it is a international project. 
  2. One major issue of this study is that: although it is desired to evaluate the performance of three runs, there is no introduction at all of the difference among the products of early, late and final run. 
  3. Because of the comment in #2, the results in section 3 mainly focused on present the score with higher or lower values, and there is no explanation of the results. When the results are improved late or final run, it is desire to know the reason behind it. 
  4. About the data resources: please provide some info of data QC for both rain gauge data and GPM IMERG data. Do you exclude any kind of data when analyzing it? In addition, what is the error of rain gauge data?
  5.  Study methods: it is not clear how did authors deal with GPM IMERG data and compared to rain gauge data. Some equations and further explanation will be necessary. 

Minor: 

  1. Quality of the figures can be improved. Current resolution is not able to read the info of labels and some legend of color bars.
  2. Lines 139-140: you mentioned that there is no research on the accuracy of GPM IMERG satellite data in China. However, it is mentioned that Yu et al 2019 study the performance of GPM IMERG's products in the eastern coast area of China in line 66. 
  3. What is the time resolution of both rain gauge and GPM observations? How to obtain the 1d (24-h) total accumulated rainfall data in both rain gauge and GPM?  By the way, except rain gauge data, is it possible to obtain the radar-based QPE over the mainland China and do the comparison? 
  4. Lines 211-215: after introducing the four levels of daily rainfall as light rain, moderate , heavy and rainstorm rainfall, why using different threshold when examining the GPM IMERG data?  
  5.  There are several sentences keep repeating in the entire manuscript. For instance: sentence in lines 218-219 appears several times in the contents. 

Author Response

Thank you for your comments concerning our manuscript “Comprehensive evaluation of Global Precipitation Measurement Mission (GPM) IMERG precipitation products over mainland China”, that we submitted to water (water-1447762).

We have revised the manuscript, and would like to re-submit it for your consideration. We have addressed the comments raised by the reviewers, and the amendments are highlighted in red in the revised manuscript. Point by point responses to the reviewers’ comments are listed below this letter. We hope that the revised version of the manuscript is now acceptable for publication in your journal.

I look forward to hearing from you soon.

With best wishes,

Yours sincerely

Linjiang Nan

Round 2

Reviewer 2 Report

In the revised version, authors created a very nice section for discussions, and they modified most of the contents based on the previous suggestions / comments. However, there are still some questions and comments need to be addressed before accepting the manuscript. Since I only obtained the pdf file of tracking differences, the lines are based on that file: 

  1. Line 45-46: please also provide both advantage and disadvantage of using radar observations.
  2. Although authors have briefly introduced the three products of early, late and final runs in lines 159-167, it is still not clear why the evaluation of early, late and final products becomes research hot-spot as mentioned in lines 71-72. You may want to provide some further information of who are the potential users for "early", "late" and "final" runs. If final run is the best products, why and what is the motivation to do the inter-comparison among these three products? 
  3. My previous comment # 4, I did not see where did you include /state the information of quality control for the rain gauge and GPM IMERG data. Please provide the specific lines for the modifications. In addition, what is the quantity of random and system errors of rain gauge data? 
  4. For the study methods, I did not see further information is provided in the revised manuscript.  Lines 176-178 only briefly introduce that the IMERG data is evaluated on the point scale. But how? 
  5. Quality of the figures: Figs. 2 - 11, the x- and y- axis which provide the info of latitude and longitude are not clear at all. 
  6. Line 265: “perhaps” is not a proper word for scientific manuscript.

  7. For the results, (for instance: lines 303-307), except the density of the stations, authors may want to link the results to what are the major differences among three runs and why they could have major or minor differences. 

For the response to reviewer, please clearly state the specific lines which you modified in the revised manuscript based on the suggestions/comments, so it will be easier to track. Thank you. 

Author Response

(The authors gave the same response as above.)

Round 3

Reviewer 2 Report

Authors modified the manuscript based on the previous comments and suggestions, and I only have some suggestions which I insist that it is necessary to include / modify in the final version of the manuscript. 

  1. Line 44: please explain that what do you mean the limited utility in cold weather of weather radar.
  2.  Line 162: please briefly introduce what is the rigorous quality control of the source data. 
  3. Line 206-207: please clearly provide the information of that how to check the consistency and eliminated the "outlier" (typo in the revised manuscript) carefully in this study. 
  4. Line 209-213: please provide at least 1-2 equations to support the statements of how to do the inter-comparison between rain gauge and satellite data since the rain gauges are not equally distributed as authors mentioned in space. 
  5.  Line 337-338: Please elaborate that how the factors of terrain conditions and geographical location affect the data quality in practice. Otherwise, this is a very general and vague statements. 

Author Response

(The authors gave the same response as above.)
